# Advances in Analgosedation and Periprocedural Care for Gastrointestinal Endoscopy

**DOI:** 10.3390/life13020473

**Published:** 2023-02-08

**Authors:** Sonja Skiljic, Dino Budrovac, Ana Cicvaric, Nenad Neskovic, Slavica Kvolik

**Affiliations:** 1Faculty of Medicine, Josip Juraj Strossmayer University of Osijek, 31000 Osijek, Croatia; 2Department of Anesthesiology and Critical Care, Osijek University Hospital, 31000 Osijek, Croatia

**Keywords:** endoscopy, gastrointestinal, conscious sedation, monitoring, ambulatory patients, anesthesia, balanced, propofol, patient safety

## Abstract

The number and complexity of endoscopic gastrointestinal diagnostic and therapeutic procedures is globally increasing. Procedural analgosedation during gastrointestinal endoscopic procedures has become the gold standard of gastrointestinal endoscopies. Patient satisfaction and safety are important for the quality of the technique. Currently there are no uniform sedation guidelines and protocols for specific gastrointestinal endoscopic procedures, and there are several challenges surrounding the choice of an appropriate analgosedation technique. These include categories of patients, choice of drug, appropriate monitoring, and medical staff providing the service. The ideal analgosedation technique should enable the satisfaction of the patient, their maximum safety and, at the same time, cost-effectiveness. Although propofol is the gold standard and the most used general anesthetic for endoscopies, its use is not without risks such as pain at the injection site, respiratory depression, and hypotension. New studies are looking for alternatives to propofol, and drugs like remimazolam and ciprofol are in the focus of researchers’ interest. New monitoring techniques are also associated with them. The optimal technique of analgosedation should provide good analgesia and sedation, fast recovery, comfort for the endoscopist, patients’ safety, and will have financial benefits. The future will show whether these new drugs have succeeded in these goals.

## 1. Introduction

Gastrointestinal (GI) endoscopic interventions are performed routinely in healthcare facilities around the world. The diagnostic and therapeutic approach of these interventions is less invasive, fast, effective, and relatively simple in the hands of highly skilled medical personnel with appropriate equipment. GI endoscopies enable early detection, prevention of diseases related to the digestive system, and their early contemporary treatment. With the advancement of technology and the development of more and more complex equipment, the scope and duration of endoscopic procedures are increasing, and are provided to an increasing number of patients. During the COVID-19 pandemic, the number of GI endoscopies dropped significantly [1,2]. Therefore, it can be expected that the number of advanced diseases that will require more complex diagnostic and therapeutic approaches will increase in the following period. It is to be expected that a larger number of patients with advanced diseases, which would previously have been considered incurable, will report to the health system. These are patients with a significantly impaired condition who receive chemotherapy, biological therapy, or those who have been irradiated. They require more complex methods of monitoring and periprocedural supervision. Advances in the field of laboratory diagnostics, pharmacology, and monitoring techniques enable a personalized approach in these situations (Figure 1). This article will discuss advances and some unresolved issues related to anesthesia and periprocedural care for patients undergoing gastrointestinal endoscopies.

## 2. Type of Endoscopic Procedures and Analgosedation Procedures

In addition to relatively simple, short-term gastroscopies and colonoscopies, more complex endoscopies of the digestive tract with different diagnostic and therapeutic goals are also performed. These procedures include esophagogastroduodenoscopy (EGD) [3], endoscopic ultrasound (EUS), therapeutic endoscopic ultrasound (TEUS) [4], enteroscopy [5], percutaneous endoscopic gastrotomy [6], endoscopic removal of neoplasms, endoscopic retrograde cholangiopancreatography (ERCP) [7], and removal of foreign bodies from the GI system in children or adult patients [8,9]. The increasing role of artificial intelligence in the fields of GI endoscopic examinations such as video-capsule-endoscopy will, in the near future, enable a high level of complexity of these examinations [10]. The complexity of endoscopic examinations is determined by the urgency and duration of the procedure, as well as the general condition, age, and comorbidities of the patients undergoing it. More complex and longer endoscopic procedures are associated with greater patient discomfort and pain.

Most gastroscopies and colonoscopies are performed as part of routine screening programs. For these procedures, light to moderate analgosedation with monitoring based on clinical assessment of the level of sedation and basic vital parameters is sufficient. More complex endoscopic procedures require general anesthesia with appropriate protection of the airway, endotracheal intubation, and objective monitoring of vital parameters. In addition to the endoscopic technique itself, associated comorbidities in high-risk patients may require general anesthesia. Such conditions are obstructive sleep apnea, morbidly obese patients, procedures in patients with dementia or cognitive impairments, procedures in children, in patients with an estimated difficult airway maintenance, and various cardiorespiratory diseases.

Emergency conditions, such as active bleeding from the digestive tract with a high risk of aspiration of gastric contents, impose the need for general anesthesia and airway protection [11]. In such cases, there is an unquestionable need for educated anesthesiology staff, equipment, and objective monitoring, often with additional laboratory tests. In the case of cardiovascular and respiratory instability of a patient under general anesthesia, in addition to clinical assessment of analgosedation, electrocardiography, pulse oximetry, capnography/capnometry, and non-invasive measurement of blood pressure, there is often a need for extended monitoring. It consists of continuous measurement of systemic arterial pressure, acid-base status, and objective monitoring of the depth of sedation using the bispectral index (BIS) or entropy [12]. 

Preoperative laboratory tests should not be performed routinely. For healthy patients undergoing routine diagnostic endoscopies with normal clinical status, and who do not take any medication, no laboratory tests are necessary in preparation for anesthesia [13]. For at-risk patients, indicated laboratory tests may be selectively requested after clinical examination and history. Their goal is to correct the disorder and achieve the optimal condition of the patient before the planned anesthesia. Hemoglobin will be determined in patients with a history of anemia, the elderly, patients with a bleeding disorder, or with hematological diseases [13]. Coagulation tests will be requested before a planned endoscopy under anesthesia if the patient has a known clotting disorder, liver, or kidney dysfunction, or is taking anticoagulant therapy. This is especially important when biopsies or endoscopic resection of intraluminal polyps are planned [13].

## 3. The Level of Education of Personnel Performing Sedation Techniques in Gastrointestinal Endoscopy Practice

Procedural short-term analgosedation during gastrointestinal endoscopic interventions relieves patients’ discomfort, allowing mild muscle relaxation. Procedural sedation has become the gold standard and is a condition without which these procedures cannot be easily performed. Patient satisfaction with this service is an indicator of the overall quality of the medical service provided in healthcare institutions. Most endoscopies are short-term, outpatient, minimally invasive procedures after which the patient soon leaves the healthcare facility. The role of periprocedural analgosedation in this setting is to eliminate the anxiety, discomfort, and stress that the patient currently feels, and to ensure relaxation of skeletal muscles, thereby improving the performance of endoscopy, without affecting vital functions. Because of this, periprocedural analgosedation is widely accepted by both patients and endoscopists.

Besides, there are numerous questions and challenges surrounding the choice of the optimal drug for sedation, the medical staff that provides it, as well as the appropriate monitoring that guarantees patient safety for this type of intervention, which is why this topic is still hot in gastroenterology and anesthesiology societies [14,15]. There are no single and unequivocal guidelines based on convincing evidence and randomized studies that would result in the creation of a universal protocol and recommendations for the performance of sedation during gastrointestinal endoscopies [15].

Dossa and colleagues performed literature searches of articles published from 2005 to 2019. They have identified 19 different guidelines and seven recommendations for analgosedation techniques for performing endoscopic gastrointestinal examinations, which include the choice of different sedation drugs, the type of competent health personnel, and appropriate monitoring. Based on these results, they concluded that the existing recommendations and guidelines vary significantly depending on the countries and professional societies that issued them and that most of them are not based on strong scientific evidence [15]. Sedation techniques, the selection of drugs, monitoring, and the degree of specific education of the professional staff who provide the service during gastroenterology endoscopic examinations, differ between continents and countries around the world. In some European and Asian countries, specially trained nurses and gastroenterologists are the ones who provide sedation for endoscopic gastroenterological procedures. According to the meta-analysis by Dossa et al., the personnel providing light to moderate sedation may be educated nurses, as well as the endoscopist themselves, while it is recommended by most professional gastroenterology societies that analgosedation for complicated patients should be performed by an anesthesiologist [15]. 

The choice of the type of medication for sedation also determines the level of education of the medical staff who handle it. For example, the American Society of Anesthesiologists (ASA), the Royal College of Anaesthetists in Great Britain, and the British Society of Gastroenterology (BSG) issued a recommendation for the most used sedative/anesthetic propofol. They recommend that a healthcare provider using propofol must be specially trained in airway management [11,16].

In their study, De Paulo et al. showed that propofol administration with non-anesthesiologist-administered propofol (NAAP) in healthy ASA status I and II patients is as safe and effective as when administered by an anesthesiologist [17]. Although there are such studies, in their joint position statement the Royal College of Anaesthetists, the British Society of Gastroenterology, and the Joint Advisory Group confirmed the leading role of anesthesiologists in deeper sedation with propofol, especially in risk groups, the elderly, and patients with numerous comorbidities [18]. Despite this, its application by the NAAP is still illegal in many countries, which imposes the need for additional education of non-anesthesiology personnel according to special curricula and practical courses that should be developed. At the same time, studies show that the use of propofol in the hands of NAAP is as effective and safe for the patient as the use by an anesthesiologist and is significantly more economically acceptable [18]. Due to the possible side effects and the need to use propofol in high-risk categories of patients, the need for a safer drug has become imperative.

A survey among gastroenterologists in Canada found that more than 90% of endoscopists use sedation during endoscopic procedures, while in Europe there are large variations between individual countries. Due to possible allergies, and medico-legal obligations for airway training, both between endoscopists and anesthesiologists there is an obvious need for valid alternative [16]. For non-anesthesia personnel who should perform moderate and deep sedation, there is a clear need for additional training and education to take care of respiratory complications. Potential complications in the deeply sedated patient are aspiration pneumonia, hypotension and other cardiovascular complications [19]. The target effect of sedation is so-called “conscious sedation” and cooperative patient. This also implies that the staff can clinically assess the depth of sedation at any time, and that they have knowledge of the pharmacodynamics and pharmacokinetics of prescribed drugs [20].

Most sedation procedures require an available anesthesiology team to be available for the needs of emergency intervention, especially related to the compromised airway. Besides, some parts of the world suffer from a lack of highly educated anesthesiologic staff for short-term sedation during the performance of a large number of such examinations on a daily basis. Such a procedure is, for example, a routine or screening colonoscopy. Therefore, the cost–benefit ratio of anesthesiologic engagement for this purpose is often questioned. In contrast to the costs, patient safety must be guaranteed and is a fundamental condition of every medical service provided in controlled medical conditions according to the principle of “*primum non nocere*”. Important factors that influence the need for anesthesiologic presence are also the ASA status of the patient ASA ≥ III, Mallampati status ≥ three, the presence of anomalies/deformations of the face and airway, perioperative assessment of difficult intubation, and mask ventilation. Patients’ comorbidities, non-cooperation, chronic opioid use, and duration and complexity of the endoscopic intervention itself further complicate the decision and require a detailed periprocedural assessment and triage of the patient by a specialized physician [15].

Due to all the above, there is still a lack of a unique, ideal anesthetic/sedative that would be of use in creating applicable guidelines. An ideal sedative should have quick and simple administration, rapid onset of action, achieve anxiolysis and light to moderate sedation, possible amnesia, have desirable pharmacokinetic and pharmacodynamic properties, and enable the performance of gastroenterological procedures for ambulatory patients. At the same time, it should be free of unwanted effects on the respiratory and cardiovascular systems, and should have rapid elimination, and cessation of action after discontinuation of use, with no or negligible residual effect and rebound phenomenon. For this purpose, numerous anesthetics, sedatives, and opioid and non-opioid analgesics have been used over the years, as well as various adjuvant drugs that are mostly used in everyday anesthesiology practice.

## 4. Old and Novel Sedation Techniques for Gastrointestinal Endoscopies

Among the most used traditional drugs for sedation in gastroenterology practice are benzodiazepines in combination with opioids [16]. For example, in Great Britain, over 2.5 million endoscopic interventions are performed annually under the sedation effect of benzodiazepines and opioids, mostly prescribed by the endoscopists themselves [11]. Among benzodiazepines, the use of midazolam prevails. Midazolam is a well-studied benzodiazepine class sedative/anesthetic agent with antero-grade amnesic properties. It can be administered in combination with other drugs, especially opioids which potentiate its sedative and analgetic properties. It is preferred over other benzodiazepines because of its pharmacokinetic profile with rapid effects after and short duration of action, which makes it the favorite choice of sedation in short, non-surgical procedures, and outpatient examinations. A meta-analysis of studies comparing the effects of midazolam versus the newer remimazolam confirmed that more rescue drugs were used with midazolam, with a higher frequency of side effects, especially hypotension [21]. Remimazolam, according to current knowledge, might be a quality substitute for midazolam.

**Propofol** is currently the backbone of sedation in gastroenterological interventions and the most used intravenous anesthetic in clinical practice for procedural sedation, alone or in combination with opioids [22]. Numerous studies show the overall superiority of propofol compared to other used analgesics/sedatives despite its certain drawbacks [18] and it is currently considered the “gold standard” for the needs of short-term sedation in endoscopies of the digestive tract (Table 1).

Propofol is characterized by rapid onset and cessation of action and high extra-hepatic clearance [23]. Due to its characteristics, it is suitable both for induction to general anesthesia but also for sedation for endoscopic procedures [24]. In subanesthetic doses, it leads to sedation and anxiolysis, the effect of which is dose-dependent. Despite its many advantages, it also has certain disadvantages. Propofol has a narrow therapeutic window and is characterized by dose-dependent cardiovascular and respiratory depression, especially in frail and elderly patients [19]. In addition, an undesirable feature of propofol is pain at the application site and the possible occurrence of infection due to the lipid formulation of the drug. There are also situations in clinical practice that impose an additional need for an alternative such as hypersensitivity to soy, peanut, and/or egg proteins that are components of the drug.

In order to examine whether the total dose of propofol used for EUS can be reduced while maintaining sedation level, a group of authors examined the addition of small doses of fentanyl and ketamine to propofol. Research has found that the administration of 50 mcg of fentanyl reduces the total dose of propofol without affecting the time of awakening, while ketamine at a dose of 0.5 mg/g prolongs the time of awakening [25].

**Remimazolam tosilate (HR7056, RT)** is a new, short-acting benzodiazepine developed in China. The drug is broken down by plasmatic tissue esterases into inactive metabolites. It is an ultra-short-acting sedative with a predictable time effect, which makes it preferable for short-term procedural sedation. As of 2020, it has been approved as a short-acting sedative and intravenous anesthetic in China, South Korea, Japan, Europe, and the USA [14]. A study by Guo et al. from 2022 shows its advantage over the use of propofol in elderly patients with cardiovascular diseases [26]. In comparative analyses with propofol, remimazolam tosilate led to significantly less respiratory depression, less pain at the injection site, and better hemodynamic stability during endoscopy in elderly patients [26]. Similar results were obtained by Tan et al. in a randomized prospective study comparing propofol and RT performed on 99 patients. They confirmed that RT 0.1 or 0.2 mg/kg is a safe and effective sedative for GI endoscopies in patients older than 65 years. In combination with 0.2-g lidocaine viscous oral liquid and 0.01 mg/kg butorphanol, it leads to less hypotension than propofol 1.0–1.5 mg/kg, although patients had worse results in immediate recall, short recall, and attention tests compared to the propofol group [27].

**Table 1 life-13-00473-t001:** Results of some studies comparing propofol with other drugs used for sedation in GI endoscopies.

Drug/Dose	Comparator	Population	Targeted Effect	Result	*p*-Value
Ciprofol/0.4 mg/kg [28]	Propofol 1.5 mg/kg	Adult, 289 pts, 43.8 vs. 44.1 year	adverse drug reactions	31.3% vs. 62.8%	*p* < 0.001
			Pain on the injection site	4.9% vs. 52.4%	*p* < 0.001
Remimazolam tosilate 0.15 mg/kg + alfentanil 5 μg/kg [26]	propofol 1.5 mg/kg + alfentanil 5 μg/kg	Elderly, 82 patients, GI endoscopies ≥ 65 years	Loss of consciousness (MOAA/S score ≤ 1	20.7 ± 6.1 s vs. 13.2 ± 5.2 s,	*p* < 0.001
			Hemodynamic events	6/39 vs. 17/38,	*p* = 0.005
			Respiratory depression	2/39 vs. 9/38,	*p* = 0.026
			Pain on the injection site	0/39 vs. 5/38	*p* = 0.025
Remimazolam tosilate (2.5 mg boluses) [29]	propofol (0.5 mg/kg)	Elderly patients, GI endoscopies, (65–85)-400 patients	Hypotension	36.5% vs. 69.6%	*p* < 0.001
			Bradycardia	1.5% vs. 8.5%,	*p* < 0.001
			Respiratory depression	4.5% vs. 10.0%	*p* < 0.05
			pain at the injection site	0% vs. 12.0%,	*p* < 0.001
Remimazolam Besylate 0.2 mg/kg + alfentanyl 10 μg/Kg [30]	Propofol 1.5 mg/kg + alfentanyl 10 μg/Kg	Gastroscopy, 914 patients,	integrated pulmonary index (etCO_2_, sPO_2_; respiratory rate, pulse rate	All favorable in remimazolam group	*p* < 0.05
			MOAAS sedation depth score	In remimazolam higher than in controls	Ns.
			Awakening time	8.37 (5.74–10.63) vs. 7.08 (4.65–9.16)	<0.001
Dexmedetomidine (DEX) [31]	propofol	Meta analysis: gastrointestinal endoscopies, endoscopic submucosal dissection	Body movement during endoscopy, hypotension, Hypoxia < 90%, recovery time	No differences between groups	*p* > 0.05
			Bradycardia	DEX significantly decreased heart rate	*p* ≤ 0.0001
Computer-Assisted Propofol (CAPS) (185.7 ± 93.2 mg) + fentanyl (88.3 ± 16.9 μg) [32]	midazolam (4.8 ± 1.6 mg) & fentanyl (107.8 ± 32 μg) sedation (MF)	Low risk patients, 60.4 vs. 61.4 years, ASA I-III grade	procedural success rate	98.2% vs. 98.7%	*p* = 0.96
			-upper endoscopy		
			-colonoscopy,	98.9% vs. 98.8%	*p* = 0.59
			Recovery time	26.4 vs. 39.1’	*p* < 0.001
			Adverse events	4.1% vs. 4.0%	*p* = 0.91
Nurse-Administered Propofol Continuous Infusion Sedation (NAPCIS) [33]	1.CAPS 2. MF sedation	3331 NAPCIS vs. 3603 CAPS vs. 3809 MF Low risk patients	Effectiveness	99.1:98.8:99.0	Ns.
			Recovery time	24.8 vs. 31.7 and 52.4 min	*p* < 0.001
			Ratio of awake on recovery room admission	86.6: 82.8%: 40.8%	*p* < 0.001

Note: GI gastrointestinal, etCO_2_ end-tidal CO_2_, sPO_2_ percentage of oxygenated hemoglobin, ASA American Society of Anesthesiologists.

**Remifentanil** should be highlighted due to its unique metabolic pathway compared to other opioids. It therefore has a short-acting analgesic effect desirable for endoscopic procedures in gastroenterology. Although it does not differ pharmacodynamically from other opioids, its rapid elimination through plasma cholinesterase makes it suitable for patients with impaired liver function, which is common in gastroenterology patients [16]. The clinical effect of sedation is achieved with a bolus dose that is continued by titrating a continuous infusion depending on the patient’s weight (0.4–0.6 µg/kg/min). This may be a drawback in the practicality of drug administration and requires additional equipment. Remifentanil is used in clinical practice as an adjunct in combination with sedatives/anesthetics such as propofol, ketamine, hypnomidate, and midazolam [34,35,36].

**Oliceridin** is a newer drug from the opioid group, that activates G-protein signaling, with exceptional selectivity for μ-receptors [37]. Classical opioids such as fentanyl act on μ-opioid receptors, which are responsible for the analgesic effect, but in the sequence of activation they also act on β-arrestin receptors. These receptors are thought to be responsible for unwanted effects such as respiratory depression and gastrointestinal side effects. Oliceridine is a μ-opioid agonist that has only 14% of the β-arrestin activity compared to morphine. It was approved by the FDA in 2020 for the treatment of moderate and severe acute pain [38]. No dose adjustment of oliceridine is necessary for patients with renal impairment or for patients with mild to moderate hepatic impairment. Dose reduction is necessary in severe renal impairment [39]. Although the first data are promising, prospective randomized studies are needed to evaluate whether it will be capable of reducing the risk of GI and respiratory events compared to conventional opioids.

**Dexmedetomidine** is a sedative drug, a short-acting highly selective alpha-2 receptor agonist. It has an analgosedative effect, while preserving alertness, airway patency, protective reflexes, and spontaneous breathing. It is approved for sedation in intensive care units. During the administration of dexmedetomidine, the patient is in a state known as “conscious sedation”, which theoretically makes it the ideal choice of sedative for the needs of gastroenterological procedures. Among the unwanted side effects are the impracticality of drug administration in continuous infusion, and the possibility of severe hypotension and bradycardia associated with fast intravenous administration. In their meta-analysis comparing propofol and dexmedetomidine for achieving sedation during gastrointestinal endoscopies, Nishizawa et al. concluded that there was no difference in adverse cardiorespiratory events between propofol and dexmedetomidine, but that patient satisfaction in the propofol group was significantly higher compared to dexmedetomidine [31]. Future research is needed to demonstrate the possible superiority of dexmedetomidine over other drugs as a stand-alone or adjuvant analgesic for the needs of endoscopic procedures in gastroenterology.

Considerable efforts have so far been made in the discovery of new improved propofol derivatives based on a non-emulsion composition, which would avoid the shortcomings, and include various forms of propofol-prodrugs and their analogs. Previous research has not shown a significant advantage over the standard propofol emulsion that would justify the high production costs in the increasingly demanding and at the same time economically demanding healthcare market [40].

**Fospropofol** (2,6-diisopropylphenoxymethyl phosphate disodium salt) is a possible alternative to propofol, and unlike propofol, it is water-soluble. Fospropofol is a prodrug of propofol that is hydrolyzed on the surface of endothelial cells into propofol, and into phosphate and formaldehyde. Formaldehyde is then metabolized in the liver and in erythrocytes [41]. Clinical studies have not reported toxic concentrations of formaldehyde in serum that would lead to metabolic acidosis, vision loss, and death [41,42]. Its application is not as painful as with propofol, and the risk of bacterial contamination and hypertriglyceridemia is lower [40]. The pharmacokinetics of fospropofol takes place through a two-compartment model, in contrast to propofol, whose kinetics is observed through a three-compartment model [41]. Fospropofol has a slower onset of action and a longer recovery time than propofol because the prodrug must first be converted into the active component. Unlike propofol, fospropofol does not cause pain during application, but in most patients, it causes paresthesia and pruritus in the perianal area [43]. Due to potential respiratory complications like propofol, fospropofol is not an alternative for sedation techniques in gastroenterology [14].

**Ciprofol** is a newer intravenous anesthetic, a propofol derivative presented to the market as a promising alternative. Its chemical structure is very similar to propofol, so it has similar pharmacodynamic and pharmacokinetic properties. It is characterized by rapid onset of action, rapid recovery, and high clearance [44].

Phase one studies have shown that the use of ciprofol is safe in doses of 0.15–0.9 mg/kg with mild to moderate side effects [45]. Another study established the safety of ciprofol in doses of 0.4–0.9 mg/kg, with the same onset of action as propofol and higher potency [44]. Under equal conditions and equal concentrations, the free fraction of the drug in the aqueous phase is lower compared to propofol, resulting in reduced pain at the site of drug application. In a multicenter phase IIa and IIb clinical study, comparing different doses of ciprofol (0.4, 0.6, and 0.9 mg/kg) after a one-minute bolus, the duration of BIS < 60 was 6, 8, and 12 min, which corresponded to the absence of verbal response at six, eight and fourteen min [45]. The appropriate dose of ciprofol that allows a high success rate of colonoscopy is between 0.4 and 0.5 mg/kg. The time until the insertion of the colonoscope and the success rate were the same for, both ciprofol, at a dose of 0.5 mg/kg, and for propofol at a dose of 2 mg/kg. Recovery time was longer in the ciprofol group. Overall satisfaction with sedation/anesthesia is the same in the ciprofol and propofol groups [46]. Propofol is most often used for endoscopic procedures in a dose of 1–3 mg/kg [6,46]. The dose of ciprofol that can achieve the same effect is one quarter to one fifth of the dose of propofol, suggesting that ciprofol is 4–5 times more potent. The incidence of pain at the injection site after ciprofol application is 6.8%, while with propofol it is 50–80% [46,47]. Previous studies have shown that the use of long- and medium-chain triglyceride (LCT/MCT) propofol compared to conventional propofol significantly reduces the incidence of pain at the site of drug application [48].

A group of Chinese authors conducted a multi-center, non-inferiority trial on 289 patients to test the hypothesis that ciprofol at a dose of 0.4 mg/kg is not inferior to propofol at a dose of 1.5 mg/kg for sedation in patients undergoing gastroscopy or colonoscopy and that it has a good safety profile. The observed parameters were successfully performed endoscopy, induction time, endoscope insertion time, need to add medication, time until the patient was fully awake, and the time from the end of the procedure to discharge. Adverse events observed were hypotension, bradycardia, apnea, hypoxemia, hyperbilirubinemia, and prolonged QT interval. The results showed that ciprofol at a dose of 0.4 mg/kg was not inferior to propofol at a dose of 1.5 mg/kg, and that there was a lower incidence of pain at the site of drug administration compared to propofol (4.9% vs. 52.4%, *p* < 0.001) [28].

## 5. Monitored Anesthesia Care for Gastrointestinal Endoscopy

The frequency and complexity of endoscopic GI diagnostic and therapeutic procedures are constantly increasing. The purpose of monitored anesthesia care (MAC) itself is to provide patients with comfort and, above all, pain control and safety. Considering that these procedures are unpleasant and painful for the patient, they are increasingly looking for different techniques of anesthesia or sedation [49,50,51]. A gastroenterologist who performs the procedure in this way can also improve the quality and efficiency of the examination itself, because the patient is cooperative, and the gastroenterologist can be focused on performing the endoscopy [51]. Preparation for MAC is very similar to that for general or regional anesthesia. The goal is to inform the patient about the entire planned procedure and progress, identify medical risks, and optimize the patient’s condition. During the preoperative evaluation, a physical examination should be performed, and the patient’s history should be taken, which includes information about the current problem, associated diseases, information about previous anesthetic procedures, current medications being taken, allergies and potential respiratory tract disorders. In patients who are at risk of malabsorption and malnutrition, the anesthesiologist should have insight into the results of relevant laboratory findings and associated tests. The most common complications in patients undergoing gastrointestinal endoscopic procedures are hypotension, aspiration, and hypoxemia [52]. The goal of the evaluation is to identify the patient’s current condition and to create an anesthesia plan that minimizes risks. 

New means of communication, such as smartphones with mobile applications, can help in early recognition of deterioration or recovery of a patient with GI diseases. A mobile application was created for patients with inflammatory bowel disease [53]. Patients filled out a questionnaire every day, and the data were processed in the electronic medical record. Based on their answers, alerts or red flags were created, which can be monitored. The central processing of these answers can indicate the urgency of a visit to the doctor or time in which it is possible to perform the planned endoscopy and reduce the scope of examinations. Their application can help in confirming disease remission, determining the patient’s general condition, and the need for emergency or control endoscopy can be facilitated [53]. If they are at special risk, such patients will be referred to an anesthesiologist for endoscopy, and for patient safety, NAAP or other procedures performed by non-anesthesiologists should be avoided.

For MAC, similar drugs are used as for general anesthesia, in lower, sedation doses, to maintain spontaneous ventilation [54]. Therefore, continuous standard monitoring of the patient must be included (certainly ECG, non-invasive pressure measurement, and monitoring of oxygenation via a pulse oximeter) along with monitoring of the patient’s airways and breathing [55,56]. Studies have shown that capnography, i.e., monitoring of exhaled carbon dioxide (CO_2_) [56], should be included if the patient is under moderate or deep sedation. Capnography is especially indicated in general anesthesia because it can be used to confirm the existence of apnea and airway obstruction more easily. Capnography can also reduce adverse events associated with respiratory depression [57,58]. Additional monitoring, such as invasive pressure measurement, can be used in long-term therapeutic endoscopic procedures, depending on the patient’s comorbidities. In the event of complications and emergencies, equipment must be available to convert to general anesthesia [59,60].

Anesthetic procedures during gastrointestinal endoscopies may be associated with a higher risk of complications for several reasons. One of them is certainly the fact of the complexity of the procedure, which is performed outside the “classic” operating room [60]. Other risk factors include obesity, male gender, and ASA status greater than III [60].

The reasons why, in some countries, sedation for gastrointestinal procedures is provided by a nurse are lower health care costs. Therefore, attempts are being made to develop methods of sedation in low-risk patients that are not applied by anesthesiologists, but by nurses, such as computer-controlled drug delivery systems, where complications may also develop [61]. It has been shown that the most common reasons for switching to general anesthesia are inability to tolerate MAC, excessive sedation, development of hypoxia or airway obstruction, and aspiration of gastric contents [62]. Because of all this, drugs used for MAC and sedation for gastrointestinal endoscopic procedures should have a rapid onset, short action, adequate titration, and rapid recovery. Most drugs can cause respiratory depression and hypotension, and accordingly, the dose needs to be adjusted, depending on the patient. Hypoxia most often occurs with the use of propofol, and the assumption is that the cause could be its pharmacological diversity and variability. Likewise, if the sedation is too light, it can lead to coughing or laryngospasm and eventually to hypoxia, while very deep sedation leads to apnea and hypoxia again [62,63]. Safe sedation must be strived for. This includes monitoring the level of consciousness, an adequate visual examination of the patient, and monitoring the patient’s breathing and circulation [64]. One of the methods, which has often been used to assess the depth of sedation, especially when using propofol during endoscopies, is monitoring the bispectral index (BIS) [65].

BIS monitoring is a non-invasive method that processes EEG information, thus enabling measurement of the patient’s level of consciousness and depth of sedation. By using BIS monitoring, it is possible to reduce the applied dose of the drug, and the risk of too deep sedation, especially in elderly patients [65,66,67,68]. Considering the great heterogeneity of the conducted studies related to the use of BIS monitoring, other studies did not establish a specific clinical benefit that would include more adequate and improved oxygenation of patients, and a reduction in the risk of respiratory and circulatory complications [69,70,71].

After a gastrointestinal diagnostic or therapeutic procedure, the patient should be awake and oriented, with no signs of complications. The patient should be monitored after the procedure in the recovery or post-anesthesia care unit, where any problems that may develop can be quickly recognized and treated. Signs of a good recovery are full alertness, spontaneous breathing without oxygenation in room air, and hemodynamic stability. Safe discharge can be planned for these patients [72].

Due to all the above, and due to the recognition of adverse events, the presence of an anesthesiologist during sedation for gastrointestinal procedures is recognized as useful, as it reduces the possibility of unwanted consequences, improves patient satisfaction, and improves the patient’s post-procedural recovery [57,72]. 

## 6. Conclusions

Both the number and the complexity of gastrointestinal endoscopic procedures is constantly increasing in the world, with the increasingly important role of artificial intelligence. Periprocedural analgosedation during these procedures has become part of the standard care that increases patient satisfaction and the effectiveness of the intervention itself. Until now, there have been no common guidelines based on strong evidence regarding the choice of the ideal drug, medical staff, and monitoring during sedation for gastroenterological endoscopic procedures. Although propofol is the most used anesthetic for this purpose, in many countries there are open medico-legal questions regarding its safety in the hands of non-anesthesiologic personnel. Standard monitoring that would guarantee patients’ maximum safety during the intervention and at the same time be the most economically profitable has not been defined. Most of the effort is invested in discovering effective substitutes for propofol that would solve the problems related to its use in gastroenterology practice. The future will confirm if these efforts have been successful.

## Figures and Tables

**Figure 1 life-13-00473-f001:**
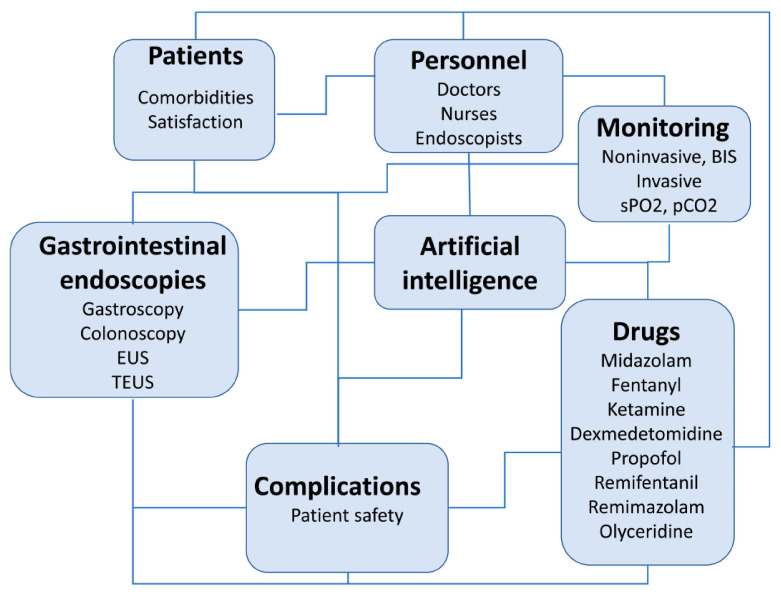
Diagram showing factors influencing periprocedural care in patients undergoing gastrointestinal endoscopies. Artificial intelligence is important in communication, monitoring, drug dosing, storing images recorded during endoscopy, and for patient safety during endoscopy. It integrates all this data and indicates caution by recognizing non-compliance with the protocol, i.e., beginning of sedation without a pulse oximeter, absence of patient or staff identification, risk factors, and deterioration of vital parameters such as hypotension or desaturation, etc.

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
