# Peer review of "Advances in Analgosedation and Periprocedural Care for Gastrointestinal Endoscopy"

_life, 2023, doi:10.3390/life13020473_

Round 1
Reviewer 1 Report
Really well written paper covering very comprehensively detailed aspects of anaesthetic practice.
It offers very up-to-date information, which will be very useful for anesthesiologists, specially in training, or indeed all staff involved in taking care of sedated of anaesthetised patients.
It is very well delivered, with very clear and easily understandable English.
I would definitely recommend to accept it to the Journal, which is actually read by above mentioned audience.
Author Response
Upon your suggestion, we carried out additional grammar corrections and spellcheck.

Reviewer 2 Report
The authors showed an overview of analgosedation for gastrointestinal endoscopy and described each sedation technique and monitoring method. This paper is well written. The current situation and issues are clearly presented. This would provide an important contribution to standardization of analgosedation for gastrointestinal endoscopy. However, a minor issue remains. The diagram of Figure 1 is difficult to understand because the relationship of each box is unclear. They had better reorganized it and clearly show the meanings of each bar connecting boxes.

Author Response
Rev. 2. However, a minor issue remains. The diagram of Figure 1 is difficult to understand because the relationship of each box is unclear. They had better reorganized it and clearly show the meanings of each bar connecting boxes.
The aim of the image was to show all the factors that are important for gastrointestinal endoscopy in one image. Considering that their interrelationships are numerous, it is difficult to make one-way relations. Thus, for example, emergency patients require additional diagnostics, and the obtained findings require a more professional team that will know how to interpret them and modify techniques accordingly. Such patients require more extensive monitoring during the endoscopy procedure, which again requires trained endoscopists who can intervene quickly in the event of complications such as bleeding, and trained anesthesiologists who, with monitoring, will be able to resolve systemic complications such as hypotension or hypoxia. Artificial intelligence integrates all this data and indicates caution by recognizing, for example, risk factors, non-compliance with the protocol (beginning of sedation without a pulse oximeter), absence of patient or staff identification, etc. It is also important in (potential) lawsuitsand internal quality controls or external audits. We added this explanation in the figure legend and reorganized the diagram accordingly.

Reviewer 3 Report
The paper submitted reviews a sensible subject regarding endoscopic procedures in present times when the comfort of patients and quality of examinations are high standards. It is true that different countries have specific legal requirements regarding sedation in endoscopy.
Regarding this article, I add more words about the role of midazolam in endoscopy since there are many places that still use it and more about analgesia and optimal drugs used for that in endoscopic procedures
I suggest adding the minimum requirements for endoscopy lab in order to be safe in using sedation
Lines 441 and 442 seems not related to the article.
Author Response
Rev. 3. Regarding this article, I add more words about the role of midazolam in endoscopy since there are many places that still use it and more about analgesia and optimal drugs used for that in endoscopic procedures.
In accordance with your suggestion, we have added some details about the use of midazolam. Considering that the aim of the article is to show advances in gastrointestinal endoscopy, we focused more on newer, short-acting drugs with fewer side effects. Remimazolam, according to current knowledge, might be a quality substitute for midazolam.
I suggest adding the minimum requirements for endoscopy lab in order to be safe in using sedation.
We added the minimum requirements for endoscopy lab ln.97-107 according to your suggestion.
Lines 441 and 442 seems not related to the article.
Thank you for pointing out that error. These two lines were removed.

Round 2
Reviewer 1 Report
I have no further comments on this paper